# Integrated Multi-Omics Signature Predicts Survival in Head and Neck Cancer

**DOI:** 10.3390/cells11162536

**Published:** 2022-08-16

**Authors:** Ilda Patrícia Ribeiro, Luísa Esteves, Francisco Caramelo, Isabel Marques Carreira, Joana Barbosa Melo

**Affiliations:** 1Cytogenetics and Genomics Laboratory, Institute of Cellular and Molecular Biology, Faculty of Medicine, University of Coimbra, 3000-548 Coimbra, Portugal; 2Coimbra Institute for Clinical and Biomedical Research (iCBR), Center of Investigation on Environment Genetics and Oncobiology (CIMAGO), Faculty of Medicine, University of Coimbra, 3000-548 Coimbra, Portugal; 3Center for Innovative Biomedicine and Biotechnology (CIBB), University of Coimbra, 3004-531 Coimbra, Portugal; 4Clinical Academic Center of Coimbra (CACC), 3000-370 Coimbra, Portugal; 5Laboratory of Biostatistics and Medical Informatics, iCBR-Faculty of Medicine, University of Coimbra, 3000-548 Coimbra, Portugal

**Keywords:** survival biomarkers, omics data, copy number alterations, methylation, gene expression, head and neck cancer

## Abstract

Head and Neck Cancer (HNC) is characterized by phenotypic, biological, and clinical heterogeneity. Despite treatment modalities, approximately half of all patients will die of the disease. Several molecular biomarkers have been investigated, but until now, without clinical translation. Here, we identified an integrative nine-gene multi-omics signature correlated with HNC patients’ survival independently of relapses or metastasis development. This prognosis multi-omic signature comprises genes mapped in the chromosomes 1q, 3p, 8q, 17q, 19p, and 19q and encompasses alterations at copy number, gene expression, and methylation. Copy number alterations in LMCD1-A1S and GRM7, the methylation status of CEACAM19, KRT17, and ST18, and the expression profile of RPL29, UBA7, FCGR2C, and RPSAP58 can predict the HNC patients’ survival. The difference higher than two years observed in the survival of HNC patients that harbor this nine-gene multi-omics signature can represent a significant step forward to improve patients’ management and guide new therapeutic targets development.

## 1. Introduction

Head and Neck Cancer (HNC) is a heterogeneous group of cancers, where more than 90% are squamous cell carcinomas arising from the epithelial mucosa at the upper aerodigestive tract [1]. Annually, HNC accounts for more than 650,000 new cases and 330,000 deaths worldwide [2]. These tumors, usually diagnosed in older patients with heavy tobacco and alcohol consumption, are facing a slow decrease globally probably due to the reduction in tobacco use [3]. Nevertheless, in developed countries, human papilloma virus (HPV)-positive oropharyngeal cancer has been increasing among younger people [3]. This neoplasm remains a substantial health and economic burden worldwide, given its high incidence and low survival rate, which have not significantly improved despite the new therapeutic strategies, such as the introduction of immune-checkpoint inhibitors and the progress in standard therapy like radiotherapy and the minimally invasive organ-sparing surgical techniques [4]. The prognosis of these patients is generally poor due to the frequent late diagnosis making therapy less effective and prone to recurrence. HNC patients with advanced disease have an average five-year survival rate of less than 50% [5,6]. The development and progression of HNC result from a multistep process of molecular accumulated alterations at different levels that compromise key cellular processes [7], namely, the accumulation of copy number alterations (CNAs), somatic mutations, and changes in methylation that consequently lead to variations in gene expression levels and downstream signaling pathways [8]. The advent of new omics technologies has allowed quantifying the number of specific molecules, such as genes, mRNA, protein, and metabolite levels of a complex and heterogeneous biological system, describing its intricate relationships and connections. In multifactorial pathologies, such as cancer, the analysis and integration of different omics layers are pivotal to deeply understanding the molecular mechanisms underlying the disease, opening new routes for a personalized diagnosis and treatment based on different molecular disease subtypes and patient stratification. Some studies, individually or combined, have decoded the molecular landscape of HNC using different omics strategies, like genomics [6,9,10,11], transcriptomics [12,13], proteomics [14,15], and DNA methylome [16,17,18], providing new insights of HNC pathophysiology and potential diagnostic and prognostic biomarkers. However, presently, little progress has been made in the translation of the molecular data to improve diagnostic and prognostic tools and to develop new therapeutic strategies for HNC patients [1]. The identification and validation of specific, robust, and integrative molecular signatures and prognosis biomarkers with clinical applicability is of utmost importance for HNC fragile patients to improve their survival and quality of life. In this study, we developed an integrative approach to analyze and combine multiple omics data, such as genomic, transcriptomic, and methylome data aiming to identify and validate a multi-omics signature related to the prognosis of HNC patients. By applying several statistical methods and machine learning approaches, we identify an integrative multi-omics signature correlated with HNC prognosis, presenting an association with patients’ survival.

## 2. Materials and Methods

### 2.1. Study Sample

The analyzed cohort is comprised of tumor tissue from 410 head and neck squamous cell carcinoma (HNSCC) patients, obtained from The Cancer Genome Atlas (TCGA). Data from copy number alterations, mRNA expression, and gene methylation were retrieved through the Genome Data Commons (GDC) Data Portal. These 410 cases were selected based on the availability of information regarding metastasis or relapse occurrence. The detailed description of the cohort in the study is presented in Table 1.

### 2.2. Statistical Analysis

#### 2.2.1. Multi-Omics HNSCC Data Reduction

Data from CNAs were available as a set of chromosomic regions, therefore, the genes contained in these regions were determined using the Bioconductor’s *Homo.sapiens* package [19]. The frequency of alteration was determined for the identified genes, which were then filtered by their percentage of alteration-only those that were altered in at least 30% of patients were kept for further analysis. Considering methylation data, a hard threshold value of 0.3 methylation level was set, and only the genes that were altered in at least 40% of patients remained. Data from mRNA expression were filtered by removing the genes with over 50% null values.

After the initial reduction step, a table (integration table) that reunites the remaining alterations from each type of omics was created. Further stages were done to reduce the amount of data that remained. The correlation between variables was determined by Pearson’s correlation coefficient, and only the variables that were correlated to another with a coefficient larger than 0.8, indicating a strong correlation, were kept in the integration matrix. Variables that have weak correlation do not constitute a construct and are excluded because only robust components are to be kept.

#### 2.2.2. Principal Components Analysis and Survival Analysis

A principal component analysis (PCA) was performed using the selected genes to reduce the number of variables and analyze the possible relationships between them. Afterwards, using the scores obtained for the first ten principal components (PCs), a clustering analysis was performed by a k-means method with a k++ initialization, using the software Orange Data Mining [20]. The two resulting clusters were assessed at the clinical level by performing survival analysis using the Kaplan-Meier method.

#### 2.2.3. Classification Using the First Ten PCs as Predictors for Survival

Aiming to test the hypothesis that the first ten principal components (PCs) contained information that allowed to distinguish between the two clusters found, a classification algorithm based on a Random Forest (RF) method was applied to distinguish the two groups using the scores determined for first ten PCs as features. The performance of the model was evaluated using a Monte-Carlo method with 5000 iterations (9).

#### 2.2.4. Selection of Genes for the Multi-Omics Signature

After establishing that the first ten PCs determined by PCA seemed to contain enough information to distinguish the two groups with different survival, the variable loadings were used to select the most contributing genetic regions. The loadings can be understood as weights of the genetic regions that contribute to the principal components. Therefore, the genetic alterations that contributed exclusively to each one of the first ten PCs were found by firstly selecting the alterations in the genes whose loadings were in the percentile 90 and, secondly, finding the ones that were present exclusively in one of the ten PCs.

For the RF model, the most important genetic alterations were selected using the Gini’s coefficient that was computed with a Variable Importance Plot. The bootstrapping algorithm was run 5000 times, and all variables with importance below 10,000 were filtered out.

A LASSO regression was then performed (1000 times), and the distribution of the number of models in which a particular genetic alteration was observed was employed for the final variable selection.

#### 2.2.5. Evaluation of the Proposed Multi-Omics Signature

The prediction capacity of the multi-omics genetic signature composed of alterations present in nine genes was evaluated by applying two algorithms for statistical classification: A support vector machine (SVM) model and a random forest (RF) model, using the survival groups, identified in the clustering analysis, as the dependent variable. The performance of both models is reported for 5000 iterations (9).

A receiver operating characteristic (ROC) curve analysis was carried out for each of the genetic alterations included in the models to assess its individual separation ability. The Youden’s index was also computed from the ROC curve, aiming to establish the optimum cut-off values. In addition, a binary logistic regression was constructed using the nine-gene signature, which allows quantifying the influence of each gene in the distinction of the two groups. To obtain a reduced model, the genes that had no statistical significance were removed from the model and, in order to determine the hazard ratios (HR) for survival relating to these genes, a Cox regression model was constructed.

#### 2.2.6. Association of the Obtained Clusters with the Metastatic Status of the Patients

Fisher’s exact test was employed to evaluate how the metastatic status of the patients was related to the two clusters. Furthermore, in each of the groups defined by the metastatic status, the survival of each cluster was evaluated through the Kaplan-Meier method.

All analyses were performed using R version 4.03, IBM^®^ SPSS^®^ Statistics version 24, and Orange Data Mining Toolbox. The significance level was established at 0.05.

## 3. Results

### 3.1. Multi-Omics HNSCC Data Reduction

Data from CNAs, mRNA expression, and gene methylation were retrieved from TCGA. After the initial data reduction step, 2193 genes were kept from the CNA data, 17,684 from the RNASeq dataset, and 13,043 genes resulted from the methylation data, summing a total of variables in the integration table.

Given the sheer amount of data, further reduction steps were carried out. Firstly, correlation analysis was performed, and only those variables that correlated to another, with a Pearson correlation coefficient larger than 0.8, were considered in further analysis. This resulted in a reduction to 8247 variables (Figure 1).

### 3.2. Principal Components Analysis and Survival Analysis

A principal components analysis was performed on the reduced integration table. A clustering algorithm was applied to the first ten principal components’ scores, resulting in two clusters with good separation (Figure 2) evaluated by the silhouette coefficient. The average silhouette coefficient is 0.533, which indicates good separation between clusters and good cohesion within each cluster. Cluster 1 includes 240 (58%) patients and Cluster 2 is composed of 170 patients (42%).

Survival analysis was performed using the Kaplan-Meier method to estimate the differences in the survival of the patients belonging to each of the clusters (Figure 3).

Survival analysis was performed using the Kaplan-Meier method to estimate the differences in the survival of the patients belonging to each of the clusters (Figure 3). In median, patients that belonged to Cluster 1 survived 836 days (approximately 2 years and 4 months) longer than those in Cluster 2 (Table 2). Log-rank, Breslow and Tarone tests did not show significant differences between the Kaplan-Meier curves (p_Log Rank_ = 0.181, p_Breslow_ = 0.615 and p_Tarone_ = 0.377, respectively), since for lower survival times, there is an overlap of the survival curves, which may be due to shorter follow-up times. However, since the median survival values indicate a clinically relevant difference (>2 years) between the two clusters, survival was considered as the clinical prognostic variable in the subsequent analysis.

### 3.3. Classification Using the First Ten PCs as Predictors for Survival

Having associated the survival of the patients with the two clusters, a classification algorithm based on the Random Forest method was applied to the first ten principal components, using the clusters as the dependent variable. The goal of establishing the classification model was twofold: Obtain a way of predicting if a patient belongs to the cluster with greater survival time or the other, and determine which genes are associated with this distinction. According to the statistics obtained by cluster analysis, the groups have good separability and the first ten PCs are good predictors of the clusters that are related to survival (Table 3). These results (Table 3) indicate that the molecular alterations presented in the integration table and contribute exclusively to these PCs, will allow the distinction between the two groups.

### 3.4. Selection of Genes for the Multi-Omics Signature

Each PC is a linear combination of all genes. Therefore, by analyzing the loadings (weights) of each gene to a specific PC, it is possible to select those genes that contribute the most. The genes that contribute mostly to each one of the first ten PCs were determined, which resulted in 2562 variables. Then, an importance plot routine based on the Gini’s coefficient was applied to these data, and a set of 398 variables was selected by filtering out all variables with importance below 10,000.

A Lasso regression analysis was performed across 1000 runs, and the genes that were included in the largest number of models were selected. Resorting to classification techniques allowed identifying nine genes that included genetic alterations observed in all of the three selected omics. This genetic signature included the copy number alterations found in *LMCD1-A1S* (3p26.1) and *GRM7* (3p26.1) genes, the gene expression of *RPL29* (3p21.1), *UBA7*(3p21.31), *FCGR2C* (1q23.2), and *RPSAP58* (19p12) genes as well as the methylation of *CEACAM19* (19q13.31), *KRT17* (17q21.2), and *ST18* (8q11.23) genes.

### 3.5. Assessment of Prediction Value of Proposed Multi-Omics Signature

The selected nine-gene multi-omics signature was evaluated as a good predictor for the two clusters creating two different classification models-one using SVM and another using Random Forest (Table 4). The signature shows excellent prediction ability, with mean accuracies of 96% (95% CI = [0.9519; 0.9808]) and 95% (95% CI = [0.9423; 0.9712]) in the test set, respectively. In Figure 4, a heatmap portrays the differences between the nine gene-signature between both clusters, where two different omics profiles are easily distinguishable.

To evaluate the separation ability of each gene, a ROC curve analysis was performed, using the two cluster groups as the dependent variable. The area under the curve (AUC) for each of the genes along with the respective 95% confidence interval, are represented in Table 5. All genes seem to have a good individual separation capacity, especially the CNAs observed in *GMR7* and *LMCD1-AS1* genes.

By calculating the maximum of Youden’s index for each gene, an optimum cut-off point for the distinction between the two survival groups was established (Table 6).

A logistic regression model was built using the nine-gene signature, aiming to measure the significance of the variables in the prediction. The genes that were found to be non-significant (*p* > 0.05) were excluded from the logistic model, thus, only four of the initial genes were chosen to be included. The copy number alterations present in *LMCD1-AS1*, methylation status of *CEACAM19*, as well as the expression profiles of *RPL29* and *FCGR2C* were found to be statistically significant for the distinction between survival groups (Table 7).

The resulting logistic model shows an overall accuracy of 95.9%. The null model’s accuracy was 58.5%. According to the Hosmer-Lemeshow goodness of fit test, the model has a good fit for the data with a *p*-value greater than 0.05 (*p* = 0.948). When taking together the four independent variables in the logistic model, they account for 93.9% of the variance in the tumors tested (Nagelkerke R^2^ = 0.930), meaning that they explain 93% of the reason why a patient belongs to their respective survival group.

By observing the adjusted Odds Ratios (OR_adj_) determined for the model in Table 7, it can be inferred that when the value measured for a given gene (either copy number alteration, expression, or methylation levels) is below its optimum cut-off point, the odds that this patient belongs to Cluster 2 is increased. Meaning that the patients that have either log 2 copy number alteration levels above −0.150 for the gene *LMCD1-AS1*, measured genetic expression levels above 6885.475 and 33.152 for *RPL29* and *FCGR2C*, respectively, or a methylation level higher than 0.625 in *CEACAM19* have a higher probability of belonging to Cluster 1, which signifies a higher likelihood of survival for that patient.

OR_adj_ for *LMCD1-AS1* is considerably larger than the ones calculated for the other three genes included in the model. This is due to the high degree of separation for the patients above and below the optimum cut point: Essentially, all patients in Cluster 1 have log2 copy number values over the cut-off point.

The Cox regression model built with the four genes included in the logistic regression model showed statistical significance (*p* = 0.002). *RPL29* showed a significant effect upon survival time (*p* < 0.001) with HR = 0.000042. The HR for *LMCD1-AS1*, *FCGR2C* and *CEACAM19* are, respectively, −0.678, −0.000175, and 0.871, however, they are not statistically significant.

### 3.6. Association of the Obtained Clusters with Metastatic Status of the Patients

Fisher’s exact test showed no significance for the association between the metastatic status of the patients and their belonging to a given cluster (*p* = 0.428). Additionally, the survival in each of the metastasis groups was evaluated separately, with the clusters as dependent variables (Figure 5). For the group that developed metastasis the median survival time for Cluster 1 was 823 days (95% CI = [110.7; 1535.3]) and for Cluster 2 it was 606 days (95% CI = [497.18; 714.82]). In the case of the patients that did not develop metastasis during the time of follow-up, the median survival time of Cluster 1 was 4856 days (95% CI = [1131.56; 8589.44]) and, in Cluster 2 it was 2570 days (95% CI = [938.99; 4201.01]). These findings suggest that the clusters identified are not related to the metastatic status, and the clusters may comprise different mechanisms that significantly impact the survival time.

## 4. Discussion

The integration of multi-dimensional datasets seems to give better statistical and biological results than the analysis of a single molecular layer [21]. Nowadays, there are still several challenges in dealing with multi-omics integrative analyses due to the complexity of biological systems, the technological limitations, the significant amount of biological variables, and the rather reduced number of biological samples [21]. In the HNC field, little progress has been made in utilizing the multi-omics signatures to improve diagnostic tools or therapeutic interventions. Thus, the main aim of our study was to integrate different omics layers, namely genomic, epigenomic and transcriptomic data, to draw a more comprehensive view of the biological processes of head and neck carcinogenesis and consequently to identify an integrative multi-omics signature with predictive impact on patient survival. We developed a multi-omics approach based on statistical and machine learning methods using TCGA-HNC data from 410 patients, for which complete information on clinical and molecular features was available. We identified an integrative nine-gene multi-omics signature correlated with HNC patients’ survival independently of relapses or metastasis development. This prognosis multi-omics signature comprises genes mapped in the chromosomes 1q, 3p, 8q, 17q, 19p, and 19q, and encompasses alterations at copy number, gene expression and methylation. Based on this multi-omics signature that includes:(i)copy number alterations in *LMCD1-A1S* (3p26.1) and *GRM7* (3p26.1) genes;(ii)gene expression of *FCGR2C* (1q23.2), *RPL29* (3p21.1), *UBA7*(3p21.31), and *RPSAP58* (19p12);(iii)methylation of *ST18* (8q11.23), *KRT17* (17q21.2), and *CEACAM19* (19q13.31) genes,

we were able to identify HNC patients with differences in survival higher than two years and consequently with better or worse prognosis.

Additionally, each of the genes had representation in the other two omics that were not included in the model for that particular gene, and their values were in line with what was expected. For example, *LMCD1-A1S* and *GRM7* are both located in the same chromosomic region (3p26.1). We were able to determine that this region exhibited a clear predominance of loss of genetic material in relation to the amplification. By observing the expression and methylation profiles for these genes, we were also able to find that they are frequently under-expressed and hyper-methylated.

Recurrent genomic and epigenomic alterations in some specific chromosomal regions are commonly observed in HNC [6,9,10,16,22,23,24], which has already been reported in the chromosomes presented in our multi-omics signature, key genes for cancer progression. However, HNC remains a highly lethal cancer owing to the lack of validated diagnosis and prognosis biomarkers with clinical utility. To the best of our knowledge, our study is the first to report this nine gene multi-omics signature related with the patients’ survival, independently of metastasis/relapse development. The genes included in this multi-omics signature are associated with known signaling pathways that seem to be linked to cancer, namely DNA Double-Strand Break Repair, Cytoskeletal Signaling, CREB Pathway, Keratinization, Gene Expression, Metabolism of Proteins, Signaling in Gap Junctions and Protein Ubiquitylation. Our results, through the integration of three different biological levels of data, give a step forward in the understanding of the molecular basis of head and neck tumorigenesis, as well as in the identification of prognosis biomarkers, which might have a great impact on patients’ management and with direct application in the clinical routine practice.

Our study has some potential limitations, such as the fact that we are analyzing as a single tumor entity, sampling from different anatomic locations in the head and neck region, from different tumor stages, and different treatment modalities as well as we are not taking into account the presence of risk factors that could have influence in the patients’ survival. However, considering the individual effect of each of these nine genes for the prediction of patients’ survival, we verified using a logistic regression model that four genes are statistically significant. Therefore, deletion in *LMCD1-AS1*, absence of hyper methylation in *CEACAM19*, and subexpression of *RPL29* and *FCGR2C* were found to be related to unfavorable survival in these patients. Some of these genes are already related to cancer, namely, high expression of *CEACAM19* in tumor breast samples [25] and gastric cancer tissues [26]. In order to determine the influence of the variables in the prediction of survival, we performed Cox’s regression with the aim to prove the association of the variables to the distinction between the two groups, having already proven that they are distinct at the time of survival level, by using the Kaplan-Meier method. Gene *RPL29* was shown to have an adverse effect on survival, augmenting the chance of death by 23,809 times. *LMCD1-AS1*, *CEACAM19*, and *FCGR2C* showed no significance, which could be due to the sample size since they showed high HR values.

Some of the nine genes have already been associated with survival outcomes in other studies related to cancer. For example, *RPL29* has been included in a five-gene model that allows for the distinction of high- and low-risk groups with different survival outcomes in HNC patients [27]. Another study found that the overexpression of *KRT17* is associated with the proliferation and invasion and related to poor survival of non-small cell lung cancer patients [28]. In esophageal squamous cell carcinoma the upregulation of protein *KRT17* was found to be detrimental to survival [29]. Recently, the downregulation of *ST18* was associated with short event-free survival in acute myeloid leukemia [30]. Furthermore, breast cancer patients with low *UBA7* expression levels were shown to have a poor prognosis and low overall survival [31].

Further studies are needed to understand the molecular mechanism of these genes in the HNC carcinogenesis as well as to validate their role in patients’ survival, including external validation, which is lacking in our study. To the best of our knowledge, TCGA is the most thorough repository of cancer data in existence and there is a lack of equally complete publicly available databases that contain information on the three omics analyzed in this paper for the same cohort of patients.

In conclusion, we report an integrative multi-omics analysis of HNC tumors from a total of 410 patients included in the TCGA database. The analysis and integration of the complex genomic, epigenetic, and transcriptomic data reveal a multi-omics signature with genes mapped at chromosomes 1, 3, 8, 17, and 19 that together allow to differentiate the patients according to their survival and independently of metastases or relapses development. A major finding of our study is the differences in survival of HNC patients that harbor copy number alterations in *LMCD1-A1S* (3p26.1) and *GRM7* (3p26.1) genes, gene expression of *FCGR2C* (1q23.2), *RPL29* (3p21.1), *UBA7* (3p21.31), and *RPSAP58* (19p12) and methylation of *ST18* (8q11.23), *KRT17* (17q21.2), and *CEACAM19* (19q13.31), which could help in the clinical management of the patients.

## Figures and Tables

**Figure 1 cells-11-02536-f001:**
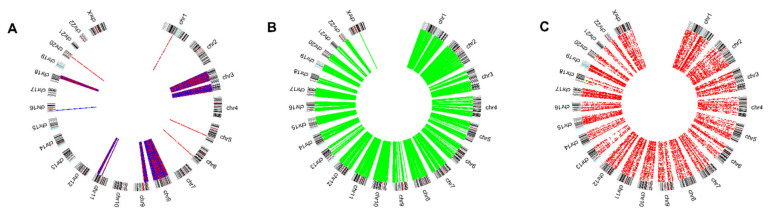
Circos plots representing the copy number alterations (**A**), methylation (**B**), and expression level (**C**) profiles of the genes kept after correlation analysis.

**Figure 2 cells-11-02536-f002:**
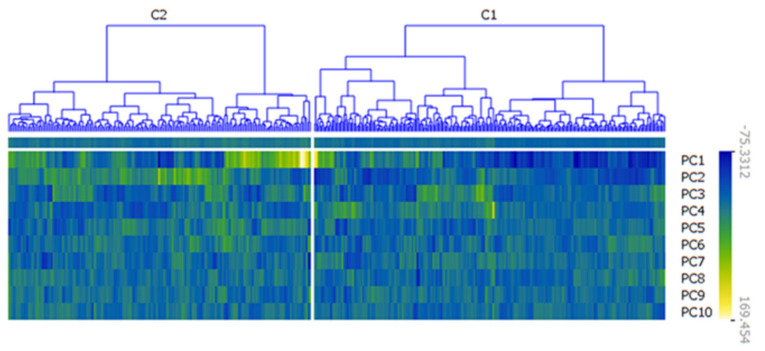
Heatmap representing the clusters obtained using the first ten principal components. Two clusters were found: Cluster 1 (**C1**) and Cluster 2 (**C2**), which, according to average silhouette coefficient (0.533), present good separation and good cohesion.

**Figure 3 cells-11-02536-f003:**
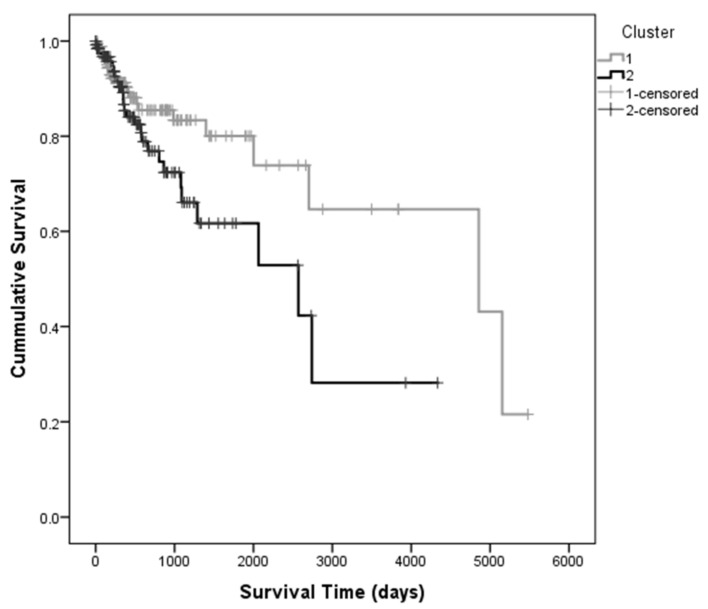
Kaplan-Meier curves determined for each cluster found using the first ten principal components of the PCA, indicating that patients in Cluster 1 survive, in median, 2 years and 4 months longer than those in Cluster 2 Patients belonging to Cluster 1are represented in grey and Cluster 2 is represented in black.

**Figure 4 cells-11-02536-f004:**
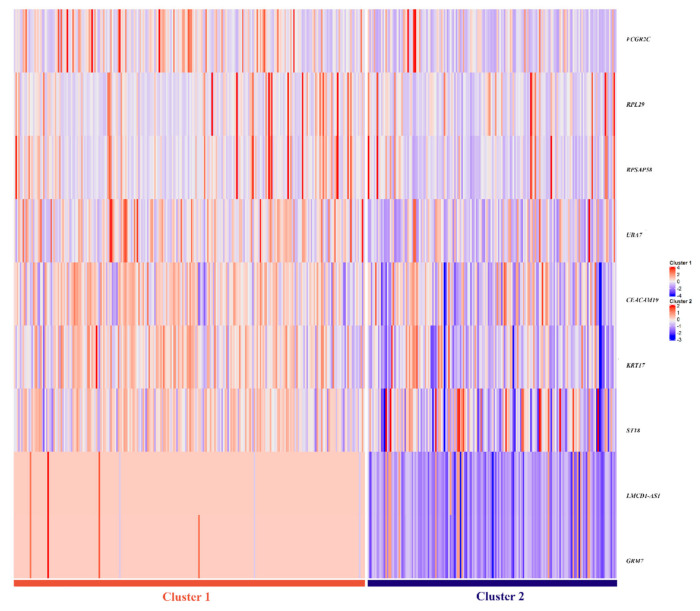
Heatmap representing the genetic profiles of the two clusters, for the nine gene signature. Lower values are represented in blue, whereas red indicates higher values. Two distinct signatures are observable, with a prevalence of lower values in Cluster 2.

**Figure 5 cells-11-02536-f005:**
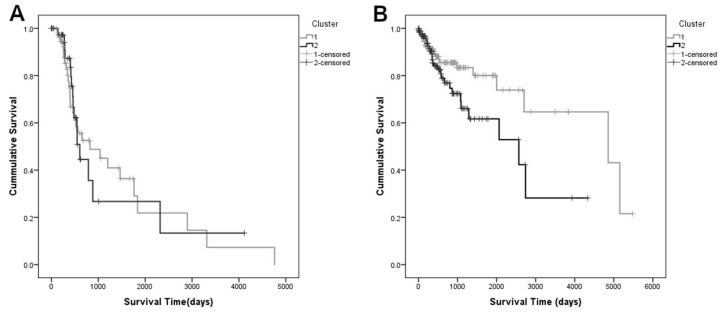
Kaplan-Meier survival curves for the group that developed metastasis (**A**) and for the group that did not present metastasis during the time of follow-up (**B**). Patients belonging to Cluster 1 are represented in grey and Cluster 2 is represented in black. Patients that did not develop metastasis or relapse in Clusters 1 and 2 present well distinguished survival profiles, as opposed to patients that did.

**Table 1 cells-11-02536-t001:** Clinic-pathologic characteristics of study population.

Patients (n = 410)			
	**n (%)**		**n (%)**
**Gender**		**HPV**	
Male	304 (74)	Positive	79 (19.5)
Female	106 (26)	Negative NA	329 (80) 2 (0.5)
**Age at diagnosis (Years)**		**Anatomic Subsite**	
<60	185 (45)	Oral Tongue Larynx Oral Cavity Floor of mouth Tonsil Base of tongue Buccal Mucosa Alveolar Ridge Hypopharynx Oropharynx Hard Palate Lip	104 (25)
≥60	225 (55)		88 (21)
**Tobacco**			54 (13)
Yes	306 (75)		45 (11)
No NA	96 (23) 8 (2)		37 (9) 23 (6)
**Alcohol**			19 (5)
Yes	280(68)		15 (4)
No	120 (29)		8 (2)
NA	10 (2)		8 (2)
**TNM stage**			7 (2)
I	19 (5)		2 (0.5)
II	72 (18)		
III	82 (20)		
IV	226 (55)		
NA	11 (3)		
**Metastasis**			
Yes	102 (25)		
No	308 (75)		

**Table 2 cells-11-02536-t002:** Means and medians for survival times for the two clusters found in the data.

	Mean				Median			
			95% Confidence Interval				95% Confidence Interval	
Cluster	Estimate	Std. Error	Lower Bound	Upper Bound	Estimate	Std. Error	Lower Bound	Upper Bound
1	2879.300	276.379	2337.598	3421.003	2900.000	702.434	1523.229	4276.771
2	2085.443	252.901	1589.757	2581.128	2064.000	652.289	785.513	3342.487
Overall	2633.589	214.999	2212.191	3054.987	2319.000	378.696	1576.757	3061.243

Survival was determined by the Kaplan-Meier method and is shown in days. Cluster 1 patients survive, in median, up to 2 years and 4 months longer than patients found in Cluster 2.

**Table 3 cells-11-02536-t003:** Evaluation metrics for test set using RF classification in the first ten PCs.

	Minimum	1st Quantile	Median	Mean	3rd Quantile	Maximum
**Accuracy**	0.8654	0.9519	0.9615	0.9624	0.9808	1.0000
**Sensitivity**	0.7500	0.9038	0.9423	0.9338	0.9615	1.0000
**Specificity**	0.8846	0.9808	1.0000	0.9910	1.0000	1.0000

Accuracy, sensitivity, and specificity are reported across 5000 runs.

**Table 4 cells-11-02536-t004:** Evaluation metrics for the test sets from SVM and RF classification using the nine gene multi-omics signature.

	Minimum		1st Quantile		Median		Mean		3rd Quantile		Maximum	
Model	SVM	RF	SVM	RF	SVM	RF	SVM	RF	SVM	RF	SVM	RF
Accuracy	0.8654	0.8365	0.9519	0.9423	0.9615	0.9519	0.9624	0.9548	0.9808	0.9712	1.0000	1.0000
Sensitivity	0.7500	0.6923	0.9038	0.9231	0.9423	0.9423	0.9338	0.9373	0.9615	0.9615	1.0000	1.0000
Specificity	0.8846	0.7500	0.9808	0.9615	1.0000	0.9808	0.9910	0.9723	1.0000	0.9808	1.0000	1.0000

Accuracy, sensitivity, and specificity are reported for 5000 runs. This signature shows high predictive ability, with the mean accuracies being 96% and 95% in the test set, respectively, for SVM and RF.

**Table 5 cells-11-02536-t005:** AUC and corresponding 95% Confidence Interval (95% CI) for the genes in the multi-omics model.

		95% CI	
Gene	AUC	Lower	Higher
*GRM7 (CNA)*	0.959	0.935	0.983
*LMCD1-AS1 (CNA)*	0.956	0.931	0.981
*RPL29 (RNASeq)*	0.658	0.605	0.711
*UBA7 (RNAseq)*	0.726	0.677	0.774
*FCGR2C (RNASeq)*	0.721	0.672	0.770
*RPSAP58 (RNASeq)*	0.717	0.667	0.767
*CEACAM19 (Methylation)*	0.723	0.674	0.772
*KRT17 (Methylation)*	0.767	0.721	0.813
*ST18 (Methylation)*	0.703	0.651	0.755

AUC: Area under the ROC curve. Good individual separation capability for patient survival is shown for the nine genes, with AUC higher than 65%.

**Table 6 cells-11-02536-t006:** Optimum cut-off points determined for each gene included in the multi-omics signature.

**Gene**	*GRM7*	*LMCD1-AS1*	*RPL29*	*RPSAP58*	*FCGR2C*	*UBA7*	*CEACAM19*	*KR17*	*ST18*
**Cut-off point**	−0.150	−0.150	6885.475	7082.378	33.152	618.547	0.625	0.344	0.573

**Table 7 cells-11-02536-t007:** Variables included in the logistic regression model and respective coefficients determined by the model.

	B	S.E.	*p*-Value	OR_adj_	95% CI OR_adj_	
					Lower	Upper
*RPL29*	2.209	0.668	0.001	9.105	2.456	33.749
*FCGR2C*	1.729	0.596	0.004	5.635	1.752	18.120
*LMCD1-AS1*	7.528	0.896	<0.001	1859.334	321.161	10,764.458
*CEACAM19*	2.668	0.812	0.001	14.405	2.935	70.707
Constant	−7.116	1.138	<0.001	0.001		

B stands for the regression coefficient, S.E. for standard error and OR_adj_ stands for the adjusted odds ratio. The copy number alterations present in *LMCD1-AS1,* methylation status of *CEACAM19* as well as the expression profiles of *RPL29* and *FCGR2C* are statistically significant for the distinction between survival groups (*p*-value < 0.05).

## Data Availability

The datasets analyzed during the current study are available in the GDC Data Portal repository, https://portal.gdc.cancer.gov/ (accessed on 19 July 2019).

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
