# Peer review of "Integrated Multi-Omics Signature Predicts Survival in Head and Neck Cancer"

_cells, 2022, doi:10.3390/cells11162536_

Round 1

Reviewer 1 Report

The manuscript entitled “Integrated Multi-Omics Signature Predicts Survival in Head and Neck Cancer” describes a nine gene signature from copy number analysis, gene expression analysis, and methylation profiling. The methods section shows clinical characteristics of the selected samples as well as the statistical analyses used in this study. The results have been presented based on the data analysis. The readers might like to know the answers for some of the questions. For example (i) Are there any regions showed higher copy number and the genes located in those regions exhibited higher expression in patients? (ii) Are there any regions with lower copy number and the genes located in those regions identified with hypermethylation? (iii) Does all these patients diagnosed at early stage or late stage?; are there any copy number changes, gene expression or methylation profiles correlated with different stages of tumors? It would be nice to show some molecular evidence from literature for these nine genes if they were association with cancer pathogenesis. In this manuscript, most of the work is related to data analysis rather than lab experiments. The authors may show experimental evidence for one of the nine genes on how they were associated with cancer pathogenesis or patient survival. In summary, the authors showed interesting results on nine genes signature by using multi-omics data analysis. However, if there is any possibility and have enough resources to perform the experiments then the authors need to address the above questions.  

Author Response

Authors: Thank you very much for your comments. We addressed each of the concerns of the reviewer individually.

Author’s response: Since points (i) and (ii) are connected we decided to offer a joint response to both questions. Firstly, we would like to thank the reviewer for their very pertinent questions. The genes included in our model that pertain to copy number alterations (GRM7 and LMCD1-A1S) are both located in the same chromosomic region (3p26.1). For the genes located in this region we were able to determine that this region exhibited a clear predominance of loss of genetic material in relation to the amplification. By observing the expression and methylation profiles for these genes, we were also able to find that they are frequently under-expressed and hyper-methylated. We also determined the same information for the other seven genes included in the model. CEACAM19, KRT17 and ST18 showed a predominance of hyper methylation, under-expression and normal copy number. RPL29 showed frequent lower expression values, lower methylation and normal copy number. As for UBA7 and FCGR2C were more frequently under-expressed, with normal copy numbers and normal methylation. RPSAP58 exhibited more frequently lower expression, normal copy number and lower methylation. We have taken the reviewer’s comments into consideration and added some phrases referring to this in the discussion section of the paper.

Author’s response: (iii) This is a very interesting question that we also considered in our analysis, but did not include in the paper, given that we found no association between staging and our model. Approximately 75% of patients are diagnosed in later stages (Stages III and IV) which gravely impacts their prognosis, so we also believe this is a very significant factor to take into account. However, after having performed correlation analysis, the staging of the tumor does not seem to be correlated with any of the copy number changes, gene expression or methylation profiles that were included in the nine-gene model determined from our work. In fact, the highest absolute value of the correlation coefficients was 0.15 for the methylation changes observed in gene KRT17, which denotes a very weak correlation. 

Author’s response: We thank the reviewer for their insight into adding further bibliographical evidence of the role of the genes in carcinogenesis. We have added a paragraph in the discussion section of the paper detailing compelling evidence of the involvement of some of the genes included in the model in the survival of cancer patients.

Author’s response: We acknowledge the suggestion to perform lab experiments. This is indeed an ongoing study. We are collecting new samples and clinical follow up data in order to show experimental evidence of these nine genes association with cancer pathogenesis and patient survival. However, we think that in this stage it will be important to publish and disseminate our results in order to also allow the scientific community to try a validation of our results using different cohorts and eventually to do some improvements, which could be pivotal to prove the generalizability/applicability of these identified genes.

Reviewer 2 Report

Nowadays, there are stillseveral challenges to deal with multi-omics integrative analyses due to the complexity of biological systems, the technological limitations, the huge amount of biological variables and the rather reduced number of biological samples.  You identified an integrative nine-gene multi-omics signature correlated with HNC patients survival independently of relapses or metastasis development.  Copy number alterations in LMCD1-A1S and GRM7, the methylation status of CEACAM19,KRT17, and ST18, and the expression profile of RPL29, UBA7, FCGR2C and RPSAP58 can predict the HNC patients’ survival.these are good finds.

1:Which of these nine genes is most important? Which one is more likely  correlated with HNC patientssurvival independently of relapses or metastasis development.Next you will do a series of validation experiments

2:Cultured cells, in vitro to see whether there is any effect on cell proliferation and differentiation

Author Response

Authors: Thank you very much for your comments. We addressed each of the concerns of the reviewer individually.

Author’s response: 1.We thank the reviewer for their question. The main objective of this work was to find a multi-omics signature that correlated to the prognosis of HNC patients. And indeed, we found that the identified survival profiles are not related to the metastatic status of the patients. This means that the signature allows for the distinction of two different groups that have different survival profiles. In our model, the signature as a whole was taken into account when testing this hypothesis. That being said, we also performed a logistic regression analysis in order to measure the significance of the variables included in our multi-omics model in the prediction of survival. The copy number alterations present in LMCD1-AS1, methylation status of CEACAM19 and the expression profiles of RPL29 and FCGR2C were found to be statistically significant for the distinction between survival groups. Furthermore, in order to determine the hazard ratios (HR) for survival relating to these genes, a Cox regression model was applied which showed that RPL29 has a significant effect upon survival time (p < 0.001).

Author’s response: 2. Thank you very much for your suggestion. This is indeed an ongoing study. We are collecting new samples and clinical follow up data in order to perform experimental validations. However, we think that in this stage it will be important to publish and disseminate our results in order to also allow the scientific community to try a validation of our results using different cohorts and eventually to do some improvements, which could be pivotal to prove the generalizability/applicability of these identified genes.

Round 2

Reviewer 2 Report

no